# Seeding amplification assay with Universal Control Fluid: Standardized detection of α-synucleinopathies

Remarh Bsoul[1], Anja H. Simonsen[2], Kristian S. Frederiksen[2,3], Kirsten Svenstrup[4,5], Sara Bech[4], Lisette Salvesen[4], Anne-Mette Hejl[4], Marcello Rossi[6], Piero Parchi[6,7], Eva L. Lund[1,3], Aušrinė Areškevičiūtė[1]*

1 Danish Reference Center for Prion Diseases, Department of Pathology, Copenhagen University Hospital, Copenhagen, Denmark, 2 Danish Dementia Research Centre, Copenhagen University Hospital, Copenhagen, Denmark, 3 Department of Clinical Medicine, University of Copenhagen, Copenhagen, Denmark, 4 Department of Neurology, Bispebjerg and Frederiksberg Hospital, Copenhagen, Denmark, 5 Department of Neurology, Copenhagen University Hospital, Copenhagen, Denmark, 6 IRCCS Istituto delle Scienze Neurologiche di Bologna, Bologna, Italy, 7 Department of Biomedical and Neuromotor Sciences, University of Bologna, Bologna, Italy

* ausrine.areskeviciute@regionh.dk

## Abstract

Seeding amplification assays, specifically the Real-Time Quaking-Induced Conversion method (RT-QuIC), have shown great diagnostic potential for α-synucleinopathies. Numerous research groups have demonstrated the method's high sensitivity and specificity using cerebrospinal fluid (CSF) samples and various RT-QuIC workflows. However, establishing a uniform and stably performing RT-QuIC protocol remains challenging. To address this, we established an RT-QuIC protocol with a Universal Control Fluid (UCF), which is simple to adopt, performs stably, and allows uniform preparation of both sample and control reactions. Firstly, we adapted and established a published 48-hour RT-QuIC protocol, including the in-house production of recombinant α-synuclein (rec α-syn), and evaluated its sensitivity and specificity through a blinded screening of an 81 CSF sample cohort consisting of Parkinson's disease (PD), dementia with Lewy bodies (DLB), Alzheimer's disease, motor neuron disease, multiple system atrophy, unidentified neurodegenerative diseases, and healthy controls. Additionally, we tested all CSF samples in three volumes to determine which volume provides the best diagnostic accuracy. The established RT-QuIC performs nearly equally well with 7 μL and 15 μL added CSF, resulting in 94% and 94.5% diagnostic accuracy, respectively. Secondly, we developed a UCF solution and tested its performance with the established RT-QuIC protocol. Results indicate that UCF, used in defined volume and concentration, standardizes the preparation of both sample and control reactions without compromising the assay's diagnostic accuracy and provides a stabilizing environment for the reactions, ensuring higher reproducibility. The established RT-QuIC protocol for pathologic α-synuclein detection in PD and

**Data availability statement:** All raw data files are available from the Zenodo database (accession number(s) https://doi.org/10.5281/zenodo.15357918)

**Funding:** The author(s) received no specific funding for this work.

**Competing interests:** KSF serves as consultant and/or member of an advisory board for Novo Nordisk, Eisai/Bioarctic, Eli Lilly, Roche Diagnostics (Remuneration paid to institution) , serves (or has served) as Principal or Sub-investigator and National Coordinator on several industry-sponsored phase 2 and 3 trials (indication Alzheimer´s disease): Roche, Roche Diagnostics, Biogen, NovoNordisk, Osuka, AbbVie (Remuneration paid to institution), serves on the Scientific advisory board (and as lecturer) on the MiCog educational program (supported by an unrestricted grant from Nestlé to the European Geriatric Medicines Society, personal remuneration), speaker and educational activities for Roche, Roche Diagnostics, Eisai/Bioarctic, Eli Lilly, Novo Nordisk, Lundbeck A/S (2023, 2024, 2025) (remuneration paid to institution), BestPractice Nordic, Alzheimerforeningen, Lundbeckfonden, Folkeuni-versitetet i Emdrup, Dagens medicin (2023, 2024, 2025) (Personal remuneration), serves as Editor-in-Chief for Alzheimer´s Research & Therapy Springer – Nature (from 2024 – 2023-2024 as Associate Editor) (Personal remuneration), receives royalties from publications with Springer-Nature and Hans Reitzels Forlag and has re-ceived funding or participated in research which has received funding from the following: Alzheimer Forsknings-fonden, Parkinsonforeningen, Aase og Ejner Danielsens Fond, KID fonden, Ellen Mørch Fonden, Jascha Fond-en, C2N, Overretssagfører L. Zeuthens Mindefond, Kong Christian den Tiendes Mindefond, Rigshospitalets Forskningspulje, Innovationsfonden, A.P. Møller fonden, ERA-PERMED, IHI, Hertzfonden, Harboefonden, Grosserer F.L.Foghts Fond, Fonden for Neurologisk Forskning, DANMODIS, Beckett fonden. Remaining authors have declared that no competing interests exist.

**Abbreviation:** AD: Alzheimer's Disease; BH:Brain Homogenate; CSF: Cerebrospinal

DLB CSF samples is highly sensitive (92–96%) and specific (93–96%). Therefore, it is important that its adoption in clinical laboratories is uncomplicated and uniform. RT-QuIC with UCF simplifies, standardizes, and stabilizes the assay's performance and, thus, could be recommended as a standard protocol for accurate detection of α-synucleinopathies.

## Introduction

α-synucleinopathies are a group of neurodegenerative diseases that include Dementia with Lewy Bodies (DLB), Parkinson's Disease (PD), together referred to as Lewy body disease (LBD), and Multiple System Atrophy (MSA) [1,2]. DLB is the second most common type of dementia after Alzheimer's disease (AD) and accounts for about 4−7% of all dementia cases [3]. Currently, the worldwide prevalence of PD is estimated to be 0.1–0.2% of the global population (over 8.5 million people) [4], whereas MSA is a much rarer disease with an estimated prevalence of 0.002–0.005% [5]. Clinically, DLB presentation includes cognitive decline and visual hallucinations, PD presents with parkinsonism, and MSA with autonomic dysfunction, parkinsonism, and cerebellar ataxia. However, initial symptoms of α-synucleinopathies and other neurodegenerative diseases are often overlapping thus making early diagnosis difficult [2].

These neurological diseases are pathologically characterized by a prion-like misfolding and aggregation of the alpha-synuclein protein (αSyn) into its disease-causing strains (αSyn$^D$) [1]. In DLB, αSyn$^D$ aggregates form Lewy bodies within neurons widely distributed throughout the brain, including the neocortex, limbic system, and brainstem. In PD, these aggregates are predominantly present in neurons of the brainstem and basal forebrain, especially in the substantia nigra [6]. In contrast, MSA is characterized by αSyn$^D$ inclusions within glial cells in basal ganglia, cerebellum, and brainstem [2]. Interestingly, recent research has revealed that the αSyn$^D$ strains in LBD and MSA exhibit distinct structural and biochemical properties, contributing to disease phenotype [7,8].

Over the last decade, several seeding amplification assays (SAA) were developed and adapted for detecting αSyn$^D$ [9]. Currently, the most prominent SAA, called Real-Time Quaking Induced Conversion (RT-QuIC), allows the detection of minute amounts of αSyn$^D$ in cerebrospinal fluid (CSF) by promoting the formation of αSyn$^D$ fibrils in the presence of suitable recombinant αSyn substrate [10]. Several laboratories have demonstrated high αSyn$^D$ RT-QuIC sensitivity (85–100%) and specificity (92–100%) for DLB and PD detection using diverse chemical and technical parameters [11–15].

CSF is regarded as the gold standard sample for SAA-based detection of disease-associated proteins due to two primary reasons: the proximity of CSF to the central nervous system, which is the principal source of aggregating proteins, and the reproducibility of CSF assay results, even when different SAA protocols are employed [9], indicating that the intrinsic composition of CSF is advantageous for

Fluid; dBH: Diluted Brain Homogenate in UCF; dCSF: Diluted CSF in UCF; dPFF: Diluted Preformed Fibrils in UCF; DLB: Dementia with Lewy Bodies; DRCPD: Danish Reference Center for Prion Diseases;F:Female; HC: Healthy Controls; M: Male; MND: Motor Neuron Disease; MSA: Multiple System Atrophy; MSA-C: Multiple System Atrophy Cerebellar; MSA-P: Multiple System Atrophy Parkinsonian; NDD: Neurodegenerative Diseases Other than αSynD; n: Number; PD: Parkinson's Disease; RT-QUIC: Real-Time Quaking Induced Conversion; SAA: Seeding Amplification Assays; SD: Standard Deviation; uNDD: Unidentified Neurodegenerative Diseases; UCF: Universal Control Fluid; αSyn: Alpha-Synuclein Protein; αSynD: Disease-Causing Conformations or Strains of αSyn.

protein amplification reactions. Moreover, some published experiments described using CSF as a diluent for serial dilutions of pre-formed αSyn fibrils [16].

When RT-QuIC is tested for αSynD detection in MSA CSF, the sensitivity among laboratories varies drastically, spanning from 10–35% [17–19], and in a single study to 75% [20]. Alternatively, olfactory mucosa samples may be more suitable for αSynD detection in MSA using RT-QuIC, as they have been reported to exhibit higher sensitivity (>80%) when compared to CSF samples [21]. Currently, the most accurate αSynD detection in MSA CSF is achieved using protein misfolding cyclic amplification (PMCA), which has a reported sensitivity of 80–97% [7]. This suggests that a different SAA than RT-QuIC is needed to amplify αSynD strains present in MSA patients [6,7].

Recently, backed up by a growing body of promising SAA data, the U.S. Food & Drug Administration issued a letter of support [22] endorsing the use of CSF αSynD SAA for patient screening for their inclusion in clinical trials targeting αSynD. This indicates that SAA introduction into clinical practice is forthcoming in the very near future. It is, therefore, important that many clinical laboratories can easily adopt αSynD RT-QuIC in their practice and make it available for use in relevant patient populations. It is equally important that the diversity of SAA protocols is minimized and that protocols which are easy to implement and maintain, and which provide standardized, stable, and reproducible results are promoted [9,23].

In this paper, our primary objective is to describe the establishment and validation of an αSynD RT-QuIC protocol, published by Rossi M. et al. [12] and by Groveman B. R. et al. [16] in our laboratory. Our second objective is to describe the development and evaluation of a Uniform Control Fluid (UCF). The composition of UCF is inspired by and closely mimicking biological CSF, to retain its gold standard benefits in all RT-QuIC reactions. Thus, by enabling a standardized microenvironment for sample and control reactions, we ensure stability and reproducibility without compromising the assay's sensitivity and specificity. UCF is also providing an option for controlled reaction compositions, facilitating further efforts towards quantitative RT-QuIC applications.

## Materials and methods

The use of patients' biological samples in this study was approved and the need for consent was waived by the Scientific Ethical Committees of the Capital Region of Denmark, approval number H-20059259. Patients' data were pseudo anonymized and accessed for the research purposes in the period of May 1, 2021 and January 22, 2025.

### Brain homogenate controls

Fresh frozen frontal cortex samples were used to prepare αSynD RT-QuIC positive and negative brain homogenate (BH) controls, used to validate recombinant αSyn (rec αSyn) batches and for definition of RFU threshold. These samples were obtained during diagnostic cranial autopsies conducted at the Danish Reference Center for Prion Diseases (DRCPD). In total, three samples from patients with *post-mortem*

confirmed diagnoses of DLB and AD, as well as neuropathologically normal brain were included. The brain samples were stored at −80°C until use and were homogenized while semi-thawed.

BHs were prepared as 10% weight/volume (w/v) solutions by chemically lysing 100 mg of tissue using Tris-buffered saline (TBS) x 10 buffer at pH 7.4, supplemented with 0.5% sodium deoxycholate and 0.5% tergitol. The lysate was transferred to M-tubes™ suitable for the Dispomix™ cell dissociator (Xiril®). Mechanical homogenization was performed at 4000 rpm for 15 seconds, and the supernatant was cleared by centrifugation at 800 g for 5 minutes at 4°C. The resulting supernatant was aliquoted and stored at −80°C until use.

## Cerebrospinal fluid samples

CSF samples were collected from a cohort of 81 individuals, comprising various clinically confirmed neurological conditions and healthy controls. The cohort included 25 samples from patients diagnosed with PD (n = 13) and DLB (n = 12), 11 samples from patients with MSA (Parkinsonian, n = 9; Cerebellar, n = 2), 12 samples from patients with AD, 11 samples from patients with Motor Neuron Disease (MND), 7 samples from patients with unidentified neurodegenerative diseases (uNDD), and 15 samples from healthy controls (HC). All CSF samples were obtained following standard lumbar puncture procedures and were stored at −80°C until use. The overview of CSF samples' cohort, including patients' sex and age at sample collection, is presented in Table 1. Twelve HC samples were provided by the Department of Pathology, Copenhagen University Hospital, while all the rest of the cohort samples were provided by the Danish Dementia Research Centre, Copenhagen University Hospital, and the Department of Neurology, Bispebjerg and Frederiksberg Hospital, Denmark.

## αSyn$^D$ RT-QuIC workflow

**Recombinant αSyn production and validation.** Recombinant histidine-tagged full-length human αSyn (rec αSyn, 1–140 aa) was designed using the nucleotide sequence of the wild-type human *SNCA* gene with NCBI accession number NP_000336.1. The N-terminal 6xhistidine-tag had a linker sequence described by Rossi et al. [12]. The gene insert was commercially (Azenta®) ligated into the PRSET (Invitrogen®) vector system and transformed into BL21(E3) Rosetta (Novagen®) *Escherichia coli* with ampicillin resistance. A 5 mL liquid culture of Lysogeny Broth media was inoculated and cultivated until an OD600 of 0.3–0.5 was reached. The culture was then diluted 1:1000 in Overnight Express Terrific Broth (OETB™, Novagen®) media and cultivated in 2 L baffled flasks at 30°C and 180 rpm overnight. Cells were pelleted at 4000 g for 15 minutes and resuspended in 25 mL osmotic shock buffer (40% sucrose, 2 mM EDTA, 30 mM Tris, pH 7.2) per 250 mL culture media. The suspension was stirred in rotation for 10 minutes at room temperature and pelleted at 7000 g for 30 minutes at 20°C. The supernatant was discarded, and the wet pellet was

**Table 1. Overview of sample cohort.**

| Clinical diagnosis | Samples (n) | Sex (M/F) | Age (median (min; max)) |
|---|---|---|---|
| DLB | 12 | 8/ 4 | 73 (69; 82) |
| PD | 13 | 11/ 2 | 72 (41; 82) |
| MSA* | 11 | 5/ 6 | 64 (57; 72) |
| AD | 12 | 7/ 5 | 73 (58; 86) |
| MND | 11 | 6/ 5 | 65 (39; 81) |
| uNDD | 7 | 3/ 4 | 64 (54; 93) |
| HC | 15 | 11/ 4 | 31 (3; 52) |

**Table 1** Overview of cerebrospinal fluid samples cohort including patients' clinical diagnosis, sex and age at sample collection. DLB – Dementia with Lewy Bodies, PD - Parkinson's Disease, MSA* - both Parkinsonian and Cerebellar Multiple System Atrophy samples, AD - Alzheimer's Disease, MND – Motor Neuron Disease, uNDD - unidentified neurodegenerative diseases, HC – healthy controls, n – number, M – male, F – female.

resuspended in ice-cold demineralized water at a ratio of 10 mL per 250 mL culture media. The resuspension was sonicated on ice (Sonifier SFX150, microtip) at 70% intensity for 30 seconds over two rounds with 10 second pause in between. The resuspension was then stirred in rotation for 10 minutes at 4°C, and pelleted at 7000 g for 20 minutes at 4°C. The supernatant was transferred to a suitable beaker and acidified to ~pH 3.5 by dropwise addition of 1 M HCl. The solution and resulting precipitate were centrifuged at 7000 g for 30 minutes at 4°C, and the pH was elevated to ~7.5 by dropwise addition of 1 M NaOH. The solution was sterile filtered (0.45 μm) and loaded (20 mM imidazole, 20 mM Tris at pH 7.5) onto a 5 mL prepacked HisTrap™ HP column (Cytiva®). The column was washed with 50 mM imidazole at pH 7.5. The protein was eluted in a linear gradient with 15 column volumes (CV) of 20 mM Tris, 500 mM imidazole at pH 7.5. The eluate (15–20 mL) was loaded onto a 5 mL prepacked HiTrap™ Q (Cytiva®) and washed with 12 CV of 20 mM Tris, 100 mM NaCl at pH 7.5. The product was finally eluted in a linear gradient with 20 CV of 20 mM Tris, 500 mM NaCl at pH 7.5. The product was sterile filtered (0.22 μm) and dialyzed overnight in 40 mM sodium phosphate buffer at pH 8. The dialysis buffer was exchanged, and dialysis continued for additional 4 hours. Protein concentration was estimated using Qubit® protein assay kit (ThermoFisher). The protein solution was aliquoted into 1 mg portions, flash-frozen in liquid nitrogen, and stored at −80°C.

The presence and purity of the rec αSyn were verified by sodium dodecyl sulfate (SDS) polyacrylamide gel electrophoresis (10% BIS-TRIS, 1 mm thick wells), followed by Coomassie staining (2.4 g/L brilliant blue, 60% methanol, 12% glacial acetic acid) and counterstaining (20% methanol, 20% glacial acetic acid).

The validation of each rec αSyn batch was based on the performance of 24 replicates containing only 98 μL master mix (referred to as unseeded or negative control), and 24 replicates with 2 μL DLB BH diluted in DEPC-water to $10^{-6}$ w/v added to 98 μL master mix (referred to as seeded or positive control). A rec αSyn batch performance was considered satisfactory if 22 out of 24 unseeded master mix replicates yielded negative results (>90%) and 23 out of 24 DLB BH seeded master mix replicates yielded positive results (>95%). A detailed description regarding the definition of a threshold for positive and negative reaction outcomes is provided in the "Interpretation of αSyn[D] RT-QuIC results" section below.

**αSyn[D] RT-QuIC.** RT-QuIC was performed using a FLUOStar Omega instrument (BMG LABTECH®) and black 96-well plates with a clear flat bottom. Each well was loaded with six 800-micron silica beads (OPSDiagnostics®) using a bead dispenser (MolGen®). RT-QuIC reactions were then added, and the plate was sealed with an optical adhesive film (MicroAmp™, ThermoFisher®). RT-QuIC script was initiated, commencing a 48-hour assay at $42°C$ with alternating intervals of 60 seconds double orbital shaking at 400 rpm and 60 seconds resting time. Fluorescence measurements were taken every 45 minutes, expressed in Relative Fluorescence Units (RFU), with excitation at $450 \pm 10$ nm and emission at $480 \pm 10$ nm. The detection range was set to 100.000 RFU.

Each RT-QuIC plate included three types of reactions: 1) negative control (98 μL unseeded master mix), 2) positive control (98 μL master mix seeded with 2 μL DLB BH diluted to $10^{-6}$ w/v in DEPC-water), and 3) CSF test with a different CSF sample volume per reaction (96 μL, 93 μL, and 85 μL master mix with 4 μL, 7 μL, and 15 μL neat CSF sample, respectively). Additionally, we tested the performance of normal brain homogenate dilutions in water ranging from $10^{-3}$ to $10^{-6}$ to compare it with the performance of unseeded negative control solely consisting of master mix. This yielded negative results, indicating no difference between the two negative control set-ups. The master mix was prepared as a stock solution before loading the 96-well plate and consisted of sterile-filtered 40 mM phosphate buffer (pH 8.0), 170 mM NaCl, 10 μM Thioflavin T, 0.0015% SDS in 1xPBS, and 0.1 μg/μL of freshly thawed rec αSyn which had been filtered through a 100 kilodalton molecular weight cut-off spin filter (Sartorius®) at 3200 g for 10 minutes at 4 °C. All reaction types, were run in quadruplicates, referred to as replicates.

**Interpretation of αSyn[D] RT-QuIC results.** The RFU threshold was calculated as the mean RFU of 120 replicates of negative reactions, including unseeded master mix and master mix with added normal brain homogenate, when the RFU tends to be the highest – within the first 10 hours of RT-QuIC assay. Furthermore, to the calculated mean RFU, we also added 4 standard deviations (+4SD).

We considered a CSF test positive if RFU threshold was crossed in at least two out of total four sample replicates within 48 hours [12,16]. If more than two sample replicates crossed the threshold, the two best-performing, i.e., with the highest RFU value, were included in further data analysis.

**Statistical data analysis.** Raw data were exported from MARS™ software (BMG LABTECH®), and the dataset corresponding to all fluorescence measurements was transferred to a worksheet. Data analysis and illustrations were made using GraphPad® Prism 10. The kinetic development over time is illustrated as the mean RFU with 95% confidence intervals (CI). The Area Under Curve (AUC) was calculated using the software's default settings, and the AUC values were compared across groups using one-way ANOVA. Potential outliers were not removed.

## Universal Control Fluid

**Production, optimization, and validation.** To resemble natural CSF composition, UCF was composed of various salts, glucose, albumin, and gamma-globulin (as detailed in Table 2), which were added to room temperature deionized water and mixed by magnetic stirring. To optimize UCF for αSyn$^D$ RT-QuIC, two concentrations were prepared: 2xUCF and 4xUCF. The prepared UCFs were sterile filtered (0.22 µm) and stored in a sterile container at 4°C until use. All UCF components were purchased from Merck®. The shelf life of UCF without added bacterial inhibitors was estimated to be a minimum of six months.

To determine the optimal UCF volume and concentration under the established αSyn$^D$ RT-QuIC reaction conditions, we conducted a series of tests with twelve replicates each, using both 2xUCF and 4xUCF. We tested 4 µL, 7 µL, and 15 µL of (a) neat 2xUCF and 4xUCF, and (b) 2xUCF and 4xUCF seeded with DLB BH ($10^{-4}$ w/v). Then, these αSyn$^D$-unseeded and αSyn$^D$-seeded UCFs in two different concentrations were added to corresponding volumes of either 96 µL, 93 µL, or 85 µL master mix in the 96-well plates. To determine which UCF volume and concentration combination performed best, a set of criteria, listed in order of importance, was applied: 1) the highest number (%) of DLB BH seeded and unseeded reaction replicates are positive, and negative, respectively, 2) the highest average AUC value of seeded replicates with the lowest 95% CI variation, and 3) the lowest average AUC value with the lowest 95% CI variation in unseeded replicates. AUC reflects the total amount of αSyn$^D$ seeding activity detected in the sample over the duration of the assay, and therefore it is used to compare the effectiveness of different RT-QuIC reactions.

**Evaluation of UCF performance with CSF and serial dilutions of DLB BH and pre-formed αSyn fibrils.** To evaluate UCF performance, considering its properties as a sample diluent, and its overall impact on αSyn$^D$ amplification kinetics, we conducted RT-QuIC reactions consisting of 93 µL master mix and of 7 µL either (a) neat CSF, (b) 6 µL CSF and 1 µL UCF, (c) 4 µL CSF and 3 µL UCF, or (d) 2 µL CSF and 5 µL UCF. In this experiment, we tested 12 randomly

**Table 2. Universal control fluid composition and concentration.**

| Composition (mM; µg/mL) | Concentration | |
|---|---|---|
| | 2xUCF | 4xUCF |
| NaCl | 238 | 476 |
| KCl | 5 | 10 |
| NaHCO$_3$ | 52.6 | 105.2 |
| NaH$_2$PO$_4$ | 2 | 4 |
| MgCl$_2$ | 2.6 | 5.2 |
| D-glucose | 20 | 40 |
| Albumin: γ-globulin | 400: 150 | 800: 300 |

**Table 2** The composition and concentrations of Universal Control Fluid (UCF). Reagents are listed in same order as when prepared. Salts and D-glucose are in mM and the protein ratio is in µg/mL.

selected CSF samples from the PD/DLB cohort of the study. We have also conducted RT-QuIC with the same CSF/UCF ratio set-up using HC CSF to ensure UCF is not inducing rec αSyn aggregation. Each CSF/UCF mix was run in quadruple and results were interpreted as described above.

To evaluate whether UCF can be used as a diluent for preparation of DLB BH seeded positive control, we performed the BH in UCF serial dilutions. Furthermore, to estimate if pre-formed αSyn fibrils (PFF, ASF-1001–05, rPeptide, LOT: r110122ASPF) could substitute the DLB BH, we also performed serial dilutions of PFF in UCF. We aimed to test if a fixed volume of the master mix could be used in both control and sample reactions. Thus, the serial BH and PFF dilutions in UCF were added to the RT-QuIC reactions in a fixed volume, determined by the best-performing UCF volume and concentration set-up.

To evaluate how UCF compares to currently used water based BH and PFF dilutions, we also performed serial DLB BH and PFF dilutions in water. The BH and PFF dilutions in water were added in 2 μL to 98 μL master mix. The performance of the two set-ups was judged by linear regression between dilution factor and maximum RFU (maxRFU).

## Results

### αSyn$^D$ RT-QuIC threshold and rec αSyn performance

For αSyn$^D$ RT-QuIC results interpretation, a threshold for a positive replicate was calculated to be 25.000 RFU. Throughout this study, we produced, validated, and used 6 rec αSyn batches. Batch-to-batch variation was observed considering the total yield of produced rec αSyn and kinetics of DLB BH seeded RT-QuIC reactions. The yield of the 6 batches ranged from 7 to 60 mg. The kinetics of the rec αSyn amplification revealed three patterns: 1) shorter lag phase (ca. 12h) and lower (maxRFU) (<75.000), 2) longer lag phase (ca. 21h) and higher maxRFU (>75.000), and 3) longer lag phase (ca. 21h) and lower maxRFU (ca. 50.000), within the total reaction time of 48h (Fig 1). The yields of the rec αSyn batches showing different kinetic patterns was as following: Batches with the rec αSyn yield of 13 mg (C) and 15 mg (A) presented pattern one, batches with the yield of 13 mg (B) and 25 mg (E) presented pattern two, and batches with the yield of 7 mg (D) and 60 mg (F) presented pattern three. Thus, indicating no correlation between the yield of rec αSyn purification and the kinetics of seeded reactions.

All batches exceeded the set quality threshold of >90% negative replicates for unseeded and >95% positive replicates for DLB BH seeded master mix reactions. The number of negative unseeded replicates in different batches ranged from 92% (22/24) to 100% (24/24), and the number of positive DLB BH seeded replicates ranged from 96% (23/24) to 100% (24/24). Three out of 6 batches had 100% negative unseeded replicates, and 4 out of 6 batches had 100% of DLB BH seeded positive replicates. Across all batches, 4% (6/144) of all unseeded replicates were false positive, and 1.4% (2/144)

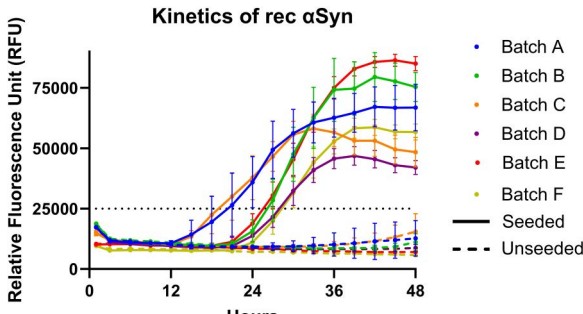

**Fig 1. Kinetics of rec αSyn batches used in this study.** Fig 1 *Kinetics of the 6 different in-house produced rec αSyn batches. Each batch was tested with 24 unseeded (dotted lines indicating mean RFU and 95% CI) and 24 seeded (continuous lines indicating mean RFU and 95% CI) reactions. Each batch is labeled with a letter (A to F) and color coded as listed in the figure legend.*

of all DLB BH seeded replicates were false negative. Although the difference between the false positive and false negative replicates is not statistically significant with the given number of total replicates, it could indicate a potential tendency of seeded RT-QuIC reactions to be more stable as compared to unseeded reactions solely composed of master mix. Together, these data indicate that despite variations in rec αSyn yield and amplification kinetics between batches, they perform equally well when the batch quality threshold is met.

## αSyn$^D$ RT-QuIC sensitivity and specificity

The sensitivity, specificity and the overall αSyn$^D$ RT-QuIC diagnostic accuracy were established based on evaluation of 81 CSF samples (Table 1) using the set criteria for a positive and negative RT-QuIC outcome. These results varied based on patients' clinical diagnosis and CSF volume added to the reaction, except for DLB and HC sample groups (Table 3, Fig 2). αSyn$^D$ RT-QuIC sensitivity for DLB, and specificity for HC remained 100% across all tested CSF volumes.

**Table 3. αSynD RT-QuIC sensitivity, specificity, and diagnostic accuracy based on 81 CSF sample cohort.**

| CSF Vol. | Sensitivity (%(CI95)) | | | | Specificity (%(CI95)) | | Diagnostic accuracy (%(CI95)) | |
| --- | --- | --- | --- | --- | --- | --- | --- | --- |
| | DLB | PD | DLB-PD | MSA* | HC | HC-NDD# | DLB-PD vs HC | DLB-PD vs HC-NDD# |
| 4 µL | 100 (76-100) | 77 (50-92) | 88 (70-96) | 0 (0-26) | 100 (80-100) | 96 (85-99) | 94 (18-100) | 92 (17-100) |
| 7 µL | 100 (76-100) | 85 (58-96) | 92 (75-98) | 0 (0-26) | 100 (80-100) | 96 (85-99) | 96 (19-100) | 94 (18-100) |
| 15 µL | 100 (76-100) | 92 (67-99) | 96 (80-99) | 9 (2-38) | 100 (80-100) | 93 (82-98) | 98 (20-100) | 94.5 (19-100) |

**Table 3** αSyn$^D$ RT-QuIC sensitivity for DLB, PD, DLB and PD together, and MSA; Specificity for HC, and HC and NDD# together; Diagnostic accuracy for DLB and PD versus HC, and DLB and PD versus HC and NDD# using 4, 7 and 15 µL cerebrospinal fluid (CSF vol.) from the 81 sample cohort summarized in Table 1. DLB – Dementia with Lewy Bodies; PD - Parkinson's Disease; MSA* - both Parkinsonian and Cerebellar Multiple System Atrophy samples; NDD# - other than αSyn$^D$ neurodegenerative diseases, including AD - Alzheimer's Disease, MND – Motor Neuron Disease, and uNDD - unidentified neurodegenerative diseases; HC – healthy controls.

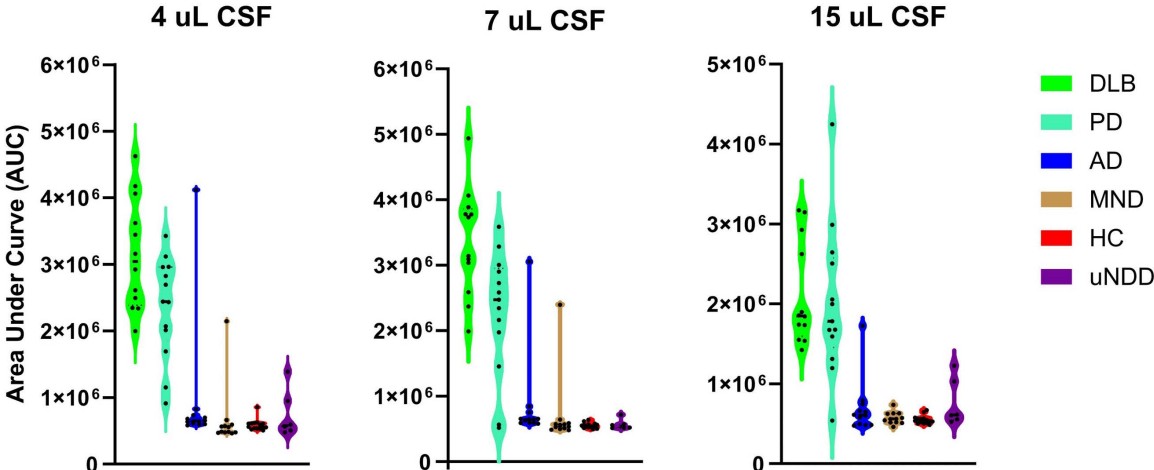

**Fig 2. Distribution of samples based on clinical diagnosis, added CSF volume and Area Under the Curve values. Fig 2** Distribution of study cohort samples (from Table 1) based on their average Area Under the Curve value (AUC) and the volume of Cerebrospinal Fluid (CSF) added to the RT-QuIC reaction. Each sample group represents a different clinical diagnosis color coded as listed in the figure legend. DLB – Dementia with Lewy Bodies; PD - Parkinson's Disease; MSA* - both Parkinsonian and Cerebellar Multiple System Atrophy samples, AD - Alzheimer's Disease, MND – Motor Neuron Disease, and uNDD - unidentified neurodegenerative diseases; HC – healthy controls.

Sensitivity for PD was lower than for DLB but increased with CSF volume increasing from 4 µL to 7 µL to 15 µL. Note-worthy, αSyn[D] RT-QuIC sensitivity for MSA was very low, namely, 0% with 4 µL and 7 µL, and only 9% (1 out of 11 samples) with 15 µL CSF.

The αSyn[D] RT-QuIC specificity decreased when HC were combined with AD, MND, and uNDD in the expected negative group. This might reflect true positive samples in this group, since dual pathology is a well-known phenomenon among elderly with neurodegenerative diseases [24].

The overall αSyn[D] RT-QuIC diagnostic accuracy, calculated as an average of sensitivity combined with specificity, increased with increasing CSF volume. When combining αSyn[D] RT-QuIC sensitivity for DLB and PD group with specificity for HC, the diagnostic accuracy increased from 94% to 96% to 98%. This demonstrates that technically, the established RT-QuIC is capable of nearly perfect DLB and PD detection and differentiation from normal controls. When combining sensitivity for DLB and PD group with specificity for HC and NDD[#] group, which is the most relevant comparison regarding αSyn[D] RT-QuIC practical adaptation in clinic, the diagnostic accuracy also increased with increasing CSF volume.

Altogether, these data indicate that the established RT-QuIC can detect and differentiate DLB and PD from a pool of different neurodegenerative diseases, including MSA, and normal controls, with 94% accuracy using 7 µL CSF, and 94.5% accuracy using 15 µL CSF.

## Universal Control Fluid properties

**Best performing UCF volume and concentration.** We determined that the best performing combination of UCF volume and its concentration, added to the RT-QuIC unseeded master mix and DLB BH seeded reactions, is 7 µL of 4xUCF. The reactions, with a 7 µL 4xUCF and 93 µL master mix set-up, were superior to other tested UCF volume and concentration combinations, when all 3 criteria in Table 4 were considered together. The 15 µL 4xUCF set-up has also performed exceptionally well, if we disregard the second criteria, the highest AUC value average of true positive replicates. Even with the markedly lower average AUC value, all the 15 µL 4xUCF seeded reaction replicates crossed the 25.000 RFU threshold. These data suggest that adding 15 µL UCF may have αSyn[D] RT-QuIC reaction suppressing effect, which could be useful for normalizing variation in seeded reaction replicates and minimizing false positive replicates in unseeded reactions (Fig 3).

**CSF diluted in UCF: Correlating AUC values and reaction kinetics.** The experiments of diluting αSyn[D] SAA positive PD/DLB CSF samples in UCF, while maintaining fixed volumes of total reaction and master mix used across all dilutions, yielded data indicating a linear correlation between total volume of added PD/DLB CSF sample and AUC value. Data show that increasing added CSF volume from 2 µL to 4 µL to 6 µL in a total dilution volume of 7 µL UCF, the AUC values are also increasing from 1.7 M to 1.9 M to 2.2 M (Fig 4a). In addition, besides the gradual decrease in RFU reflecting the serial decrease in CSF concentration, the curves presenting the kinetics of the dilution reactions remained very similar, even with only 2 µL added PD/DLB CSF (Fig 4b). This suggest that UCF presence in the reaction uniforms its

**Table 4. Overview of UCF performance.**

| Criteria for replicates | 12 replicates/ reaction | | | | | |
|---|---|---|---|---|---|---|
| | 2xUCF 4ul | 2xUCF 7ul | 2xUCF 15ul | 4xUCF 4ul | 4xUCF 7ul | 4xUCF 15ul* |
| %, true positive/ negative | 100%/ 100% | 100%/ 92% | 100%/ 92% | 100%/ 100% | 100%/ 100% | 100%/ 100% |
| Mean AUC, seeded (CI95%) | 486 (393-578) | 440 (383-496) | 384 (349-418) | 391 (344-437) | 425 (384-464) | 353 (323-384) |
| Mean AUC, unseeded (CI95%) | 148 (142-152) | 148 (140-155) | 144 (130-156) | 134 (129-138) | 131 (124-136) | 114 (111-116) |

**Table 4** *Overview of how different UCF volume and concentration combinations met the criteria for determining the best performing UCF set-up in DLB BH seeded and unseeded RT-QuIC reactions. The AUC values were divided by 10.000 for scaling. UCF – Universal Control Fluid, AUC – Area Under the Curve value. *Total number of replicates is 10 in this reaction set-up.*

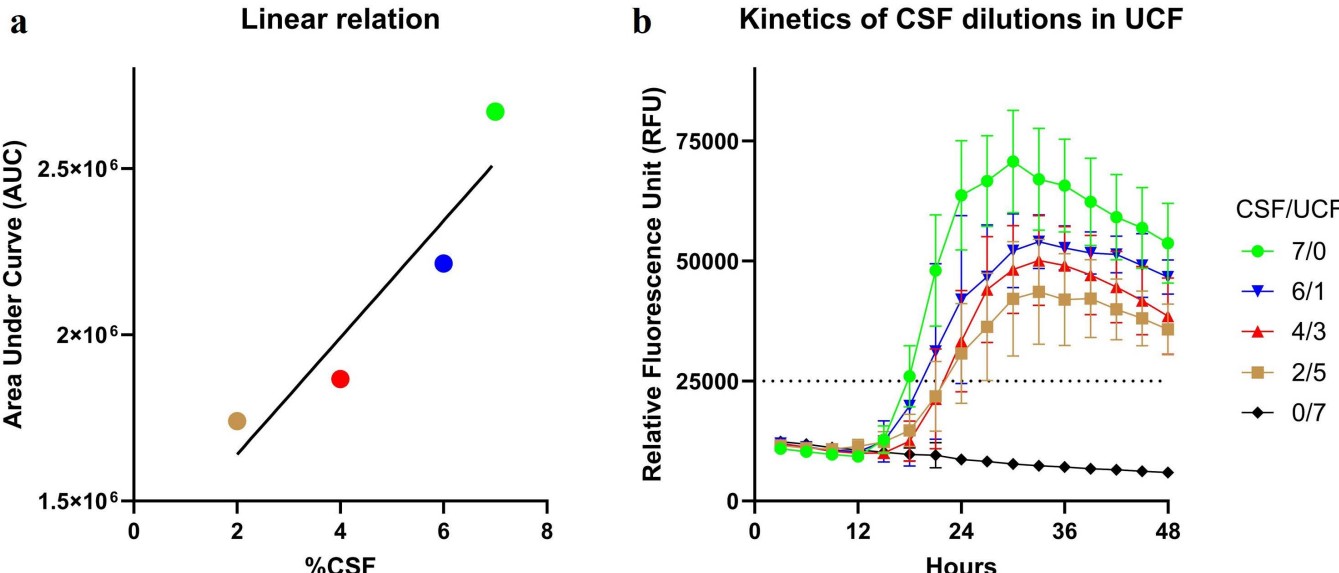

**Fig 3. BLB BH seeded and unseeded 2xUCF and 4xUCF performance.** Fig 3 *Distribution of 12 replicates based on their Area Under the Curve value (AUC) in unseeded (red) and DLB BH seeded (green) RT-QuIC reactions with different universal control fluid (UCF) volume and concentration combinations.*

**Fig 4. Linear correlation and kinetics of PD/DLB CSF dilutions in UCF and Area Under the Curve values.** Fig 4 **a)** *Linear correlation between* αSyn[D] *positive PD/DLB CSF sample dilutions in UCF and the AUC values;* **b)** *kinetics of the PD/DLB CSF samples diluted in UCF. Legend: The CSF/ UCF ratio, ranging from 7 μL neat PD/DLB CSF to 7 μL neat UCF added to* 93 μL *master mix, is color coded as indicated in the figure legend and applies to both graph* **a)** *and* **b)**. UCF – Universal Control Fluid, AUC – Area Under the Curve value.

environment across different sample dilutions added in a total volume of 7 μL, permitting a constant master mix volume (93 μL), and a standardized total reaction volume (100 μL). The same CSF/UCF ratio set-up with HC CSF, as expected, yielded negative results, thus indicating that UCF is not inducing rec αSyn aggregation in the absence of αSyn[D] in CSF.

We observed that the number of positive replicates decreased with decreasing PD/DLB CSF volume in UCF. The dilution reactions with 6 μL CSF had 87.5% of positive replicates, reactions with 4 μL CSF – 77%, and reactions with 2 μL

CSF – 71%. The same tendency of positive replicates decrease associated with decreasing added CSF volume was also observed in RT-QuIC reactions without UCF, as described in the "αSyn$^D$ RT-QuIC sensitivity and specificity" section.

**Serial dilutions in UCF: DLB BH and pre-formed αSyn fibrils.** Serial 10-fold dilutions of DLB BH in 7 µL 4xUCF with a fixed volume of 93 µL master mix, and dilutions of DLB BH in 2 µL water with a fixed volume of 98 µL master mix, spanned from $10^{-3}$ to $10^{-9}$. In case of DLB BH diluted in 7 µL 4xUCF, data indicate high linear regression ($R^2 = 0.97$) between the αSyn$^D$ dilution and RFU, showing that with increasing αSyn$^D$ concentration, RFU is also increasing (Fig 5a). The RFU threshold was crossed with all dilutions. Reaction kinetics demonstrate a smooth and gradual decrease in RFU corresponding to increasing serial dilutions, while the curves remain uniform (Fig 5b). In case of the same DLB BH diluted in 2 µL water, the linear regression is also present but lower ($R^2 = 0.84$) (Fig 5a). In addition, the $10^{-8}$ and $10^{-9}$ dilutions did not cross the RFU threshold, but most importantly, the reaction kinetics did not demonstrate a clear gradual decrease in RFU corresponding to increasing serial dilutions. Instead, the curves presenting different dilutions were grouped together (Fig 5c). These data show that the DLB BH dilutions in 7 µL 4xUCF allow $10^{-3}$-$10^{-9}$ αSyn$^D$ detection range and the consistency in amplification kinetics corresponding to a dilution factor. Thus, allowing the use of the fixed sample (7 µL) and master mix (93 µL) volumes, which also work best for CSF test reactions, making control and sample preparations more uniform. The set-up of DLB BH dilutions in 7 µL 4xUCF performed better than the set-up with water, indicating that moving away from currently used preparation of BH positive controls is acceptable.

Serial 10-fold dilutions of pre-formed αSyn fibrils (PFF) diluted in 7 µL 4xUCF with a fixed volume of 93 µL master mix, and αSyn PFF diluted in 2 µL water with a fixed volume of 98 µL master mix, spanned from $10^{-1}$ to $10^{-10}$. In case of αSyn PFF diluted in 7 µL 4xUCF (dPFF), the data indicate high linear regression ($R^2 = 0.91$) between the αSyn$^D$ dilution and RFU (Fig 6a). The RFU threshold was crossed with all dilutions and reaction kinetics demonstrate a uniform, smooth and gradual decrease in RFU corresponding to increasing serial dilutions (Fig 6b). In case of αSyn PFF diluted in 2 µL water, the linear regression is also present but lower ($R^2 = 0.66$) (Fig 6a). In addition, only the $10^{-1}$ and $10^{-2}$ dilutions cross the RFU threshold and reaction kinetics resemble those of negative replicates.

Together this data indicate that αSyn PFF diluted in UCF allow αSyn$^D$ detection range from 100 ng/ µL to 10 at/ µL, and the consistency in amplification kinetics corresponding to dilution factor. Thus, allowing PFF use as a potential alternative to DLB BH for positive control reactions. The set-up of PFF dilutions in water did not perform satisfactory regarding the set RFU threshold, and, thus, it cannot be comparable to 7 µL 4xUCF set-up.

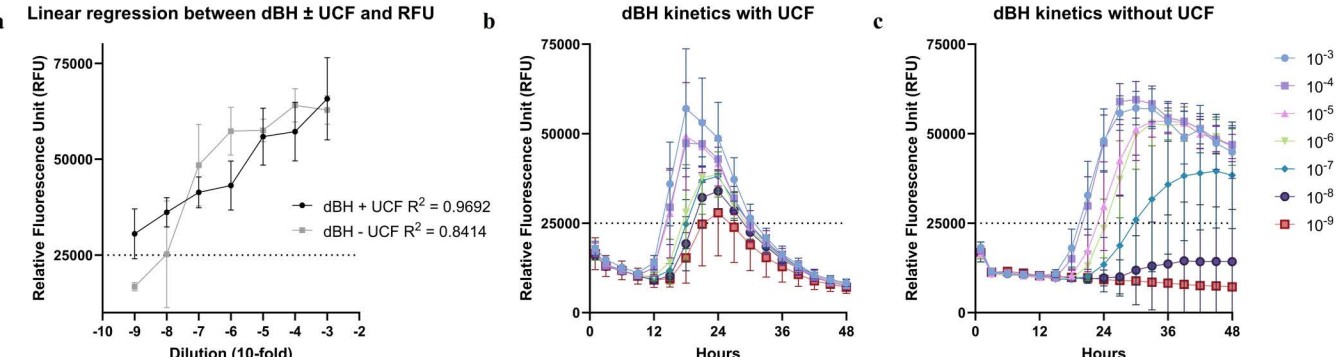

**Fig 5. Comparison of diluted DLB BH with and without UCF.** Fig 5 Comparison of DLB BH serial 10-fold dilutions in 7 µL 4xUCF and in 2 µL water. **a)** *Linear regression between DLB BH dilutions in UCF and in water, and relative fluorescence unit (RFU). The coefficient of determination ($R^2$) and color code for the comparison of linear regressions is as indicated in the figure legend;* **b)** *kinetics of DLB BH dilutions in UCF;* **c)** *kinetics of DLB BH dilutions in water.*

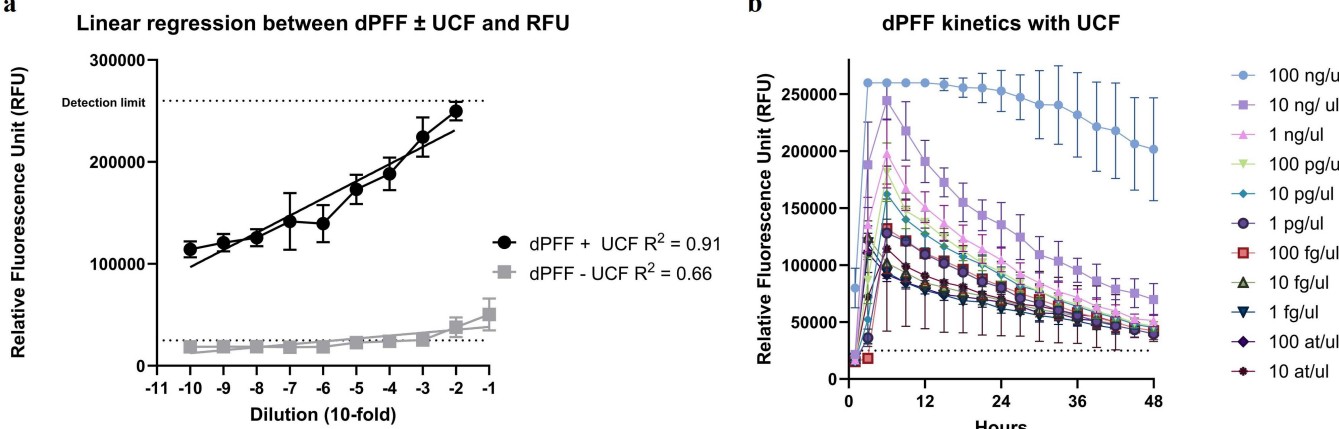

**Fig 6. Comparison of diluted PFF with and without UCF.** Fig 6 Comparison of serial 10-fold dilutions of pre-formed αSyn fibrils (dPFF) in 7 μL 4xUCF and in 2 μL water. **a)** *Linear regression between serial dilutions of pre-formed αSyn fibrils in UCF and in water, and relative fluorescence unit (RFU). The coefficient of determination (R²) and color code for the comparison of linear regressions is as indicated in the figure legend;* **b)** *kinetics of pre-formed αSyn fibrils dilutions with UCF; dilution factors and their color codes are indicated in the figure legend.*

## Discussion

In this paper, we describe a process of establishing αSyn$^D$ SAA, specifically RT-QuIC, including rec αSyn in-house production, adapting the published protocol reporting the assay's high diagnostic accuracy, short analysis time, robustness considering pre-analytical as well as analytical variables, and a potential for quantitative measurements corresponding to disease stage and clinical presentation [12,16,25–27]. In addition, we also tested and determined the impact of rec αSyn batch-to-batch differences, and the most optimal CSF sample volume added to the RT-QuIC reactions. Finally, we introduced and proved the concept of UCF, which presents an alternative to preparation of SAA controls and samples, ensuring a uniform and stabilizing reactions' environment, providing controlled composition of reactions for quantitative measurements, and, thus, standardizing detection and evaluation of α-synucleinopathies internally and across laboratories.

A reliable rec αSyn production and validation process is a key to reproducible SAA results. However, it is a multi-step process, which requires certain set of instruments and skills, it takes time, and the quality and quantity of produced batches may vary, thus posing challenges to adopting SAA in clinical laboratories. Nevertheless, throughout this study, we have realized that even without extensive prior experience in protein production, it is possible to succeed and improve your production quickly.

Moreover, the comparison of the 6 batches that we used in this study, indicated that batch-to-batch differences in rec αSyn yield, maxRFU, and amplification kinetics in αSyn$^D$ seeded reactions, do not affect the numbers of true negative and positive replicates, with a batch performance threshold set to >90% for negative, and >95% for positive replicates. We considered, a batch performance threshold lower than a 100% accuracy, to be acceptable if the controls and samples were run in quadruplicates or more replicates to effectively minimize overall false positive and negative conclusions. Therefore, in our experience, with this degree of flexibility in batch-to-batch variation, rec αSyn production is worth the effort. Additionally, it allows a full control over the components used in the RT-QuIC reactions.

Sensitivity, specificity, and the overall diagnostic accuracy of the established αSyn$^D$ SAA with the *in-house* produced rec αSyn, and a total assay time of 48 hours, was evaluated considering two aspects: DLB/PD CSF volumes added to the reaction, and comparisons of results from cohorts with different diagnoses. The results revealed that in case of DLB and HC, the assay's sensitivity and specificity remained perfectly stable, a 100%, in detecting and distinguishing DLB from HC in all tested CSF volumes. αSyn$^D$ RT-QuIC higher sensitivity for DLB versus PD has also been reported by others [16].

This could be suggestive of DLB αSyn[D] strains' ability to induce the rec αSyn conversion and aggregation more efficiently, or that higher αSyn[D] seeding activity depends on the higher Lewy body pathology load in DLB [27].

In case of PD, MSA, and other NDD, the assay's sensitivity and specificity depended on added CSF volume, with a general tendency of the higher volume improving the sensitivity but reducing the specificity. This is best exemplified in PD and other NDD cohorts, where with the increasing CSF volume, the assay's sensitivity increased from 77% to 92%, and the specificity decreased from 96% to 93%.

In case of MSA, none of the CSF samples added in 4 and 7 μL volumes yielded positive results, and only one sample (9%) became positive with 15 μL added CSF. The assay's low sensitivity for MSA, has also been reported by Rossi M. et al. using similar assay set-up [12], and could be explained by the predominant presence of a different αSyn[D] strain, which requires a different assay set-up for its amplification [28]. Nevertheless, this explanation does not exclude the possibility of isoforms' co-existence, which could explain occasional MSA detection with CSF samples tested in high volumes.

The extremely low αSyn[D] SAA sensitivity for MSA is comparable to that of negative controls cohort consisting of other NDD including AD, MND and diagnostically uncertain NDD. Thus, we deemed our αSyn[D] SAA unable to detect MSA and distinguish it from negative controls. Just like in MSA cohort, AD, MND and uNDD cohorts each had one sample that was αSyn[D] SAA positive, thus reducing otherwise perfect assay's specificity to 93–96%, depending on added CSF volume. However, this compromised specificity could be explained by the increasingly present, clinically unrecognized mixed pathologies [29–31] among included patients, who were not verified neuropathologically for αSyn[D] absence.

Nevertheless, when the overall αSyn[D] SAA diagnostic accuracy was evaluated, considering both 7 μL and 15 μL volumes, and DLB-PD cohort versus HC-NDD cohort, the difference was insignificant, i.e., 7 μL set-up performing with 94% accuracy and higher specificity, and 15 μL set-up performing with 94.5% accuracy and higher sensitivity (Table 3). Given this, it is up to individual laboratories to choose the right volume, or the combination of volumes, to be implemented in their routine diagnostics.

Currently, the SAA adaptation in clinical laboratories and diagnostic routine is challenging. The main limitation is, as described by Abdi et al. in their review, a lack of standardized and simplified protocol enabling quantitative differentiation between α-synucleinopathies [23]. As highlighted by Bernhardt et al., the inter-individual variability of the CSF can influence SAA reaction speed and kinetics and is a limitation challenging the development of quantitative SAA applications [32]. Thus, indicating a need for standardized composition of CSF test reactions. To address these limitations and enhance the αSyn[D] SAA diagnostic set-up, we developed a UCF solution, which in its composition resembles natural CSF, and can be used as a diluent to standardize the preparation of samples and controls.

Presently, in αSyn[D] SAA most often used negative control reactions consist of 98–100 μL unseeded master mix either with or without added 2 μL normal brain homogenate dilutions. We used unseeded master mix with no addition of normal brain homogenate as our standard negative control. We also tested the performance of the negative control with added normal brain homogenate and saw no difference in the performance of the two control set-ups. In other published protocols positive control reactions often consist of αSyn[D] positive BH diluted in 2 μL water and 98 μL master mix. These controls are substantially different from CSF test reactions which often consist of 15 μL pure sample and 85 μL master mix, making controls and sample reactions very different in their composition [10,12,16]. Due to these differences, positive and negative controls based on brain homogenates do not take into consideration the difference in total amount of rec αSyn and CSF composition in the test reactions.

To normalize the positive control and sample test reactions, some laboratories are using CSF samples from patients with confirmed diagnosis as an alternative to αSyn[D] positive BH controls [26]. This approach ensures uniform sample and control preparation and allows moving away from working with brain samples. It could be, however, a challenge for some laboratories to have an uninterrupted supply of αSyn[D] positive CSF samples approved for their use in diagnostics. The UCF, as a CSF diluent, could, in those situations, prolong the use of available CSF.

With the UCF we introduce a uniform preparation of negative, positive and CSF test reactions, all of which consist of fixed volumes of, respectively, 93 µL and 85 µL master mix and of 7 µL and 15 µL either CSF sample or its substitute – UCF, added to the reaction as either a neat solution for negative controls, or as a diluent for αSyn^D positive CSF, BH and preformed αSyn^D fibrils for positive controls.

Furthermore, the UCF demonstrates αSyn^D amplification suppression when added in higher volume, i.e., 15 µL, thus providing a stabilizing effect to the reactions prone to false amplification of rec αSyn. If some laboratories experience this issue, the use of 15 µL UCF and 85 µL master mix set-up for negative and positive controls could be recommended.

This αSyn^D SAA with UCF protocol ensures a uniform and simplified preparation of control and sample reactions, a 92–96% sensitivity and a 93–96% specificity for DLB and PD detection versus HC, MSA and other neurodegenerative diseases. The adoption of αSyn^D SAA with UCF protocol by other laboratories would enable multi-center clinical studies and would speed-up the development of quantitative αSyn^D SAA, and its authorization for use in diagnostics and clinical trials in the future.

## Conclusion

In this study we show that adaptation of the described αSyn^D SAA protocol in our diagnostic laboratory was a success that allows the use of both 7 µL and 15 µL CSF samples with the diagnostic accuracy of 94% and 94.5%, respectively. We also demonstrate that the development of a uniform and controlled composition of the assay reactions with the constant volumes of master mix and UCF, allows the reduction of variables that could influence αSyn^D amplification kinetics and the number of true positive and negative replicates. By standardizing the preparation of different reactions UCF could also aid the development of the quantitative SAA. The development of a standardized assay conditions is also important for multi-center clinical trials and the potential clinical diagnostic use of the assay in the future.

## Acknowledgments

The authors of this paper would like to acknowledge the Cytology Unit of the Department of Pathology, Copenhagen University Hospital, for gathering materials suitable for this study. The authors would also like to appreciate the excellent laboratory work done by biotechnicians Michelle Rasmussen and Ema R. Savelieva.

## Author contributions

**Conceptualization:** Remarh Bsoul, Kristian S. Frederiksen, Sara Bech, Eva L. Lund, Aušrinė Areškevičiūtė.

**Data curation:** Remarh Bsoul, Aušrinė Areškevičiūtė.

**Formal analysis:** Remarh Bsoul, Aušrinė Areškevičiūtė.

**Investigation:** Anja H. Simonsen, Kristian S. Frederiksen, Kirsten Svenstrup, Sara Bech, Lisette Salvesen, Anne-Mette Hejl, Eva L. Lund, Aušrinė Areškevičiūtė.

**Methodology:** Marcello Rossi, Piero Parchi, Aušrinė Areškevičiūtė.

**Resources:** Remarh Bsoul, Anja H. Simonsen, Kristian S. Frederiksen, Kirsten Svenstrup, Sara Bech, Lisette Salvesen, Anne-Mette Hejl, Eva L. Lund, Aušrinė Areškevičiūtė.

**Supervision:** Eva L. Lund, Aušrinė Areškevičiūtė.

**Visualization:** Remarh Bsoul.

**Writing – original draft:** Remarh Bsoul, Aušrinė Areškevičiūtė.

**Writing – review & editing:** Remarh Bsoul, Anja H. Simonsen, Kristian S. Frederiksen, Kirsten Svenstrup, Sara Bech, Lisette Salvesen, Anne-Mette Hejl, Marcello Rossi, Piero Parchi, Eva L. Lund, Aušrinė Areškevičiūtė.

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
