## [Decision Letter · Decision Letter 0]

Dear Dr. Areškevičiūtė,

Thank you for submitting your manuscript to PLOS ONE. After careful consideration, we feel that it has merit but does not fully meet PLOS ONE’s publication criteria as it currently stands. Therefore, we invite you to submit a revised version of the manuscript that addresses the points raised during the review process.

 Reviewer 2 was more positive. This reviewer made several points, which I am certain you can address. Please carefully address the points raised by the reviewers.

We look forward to receiving your revised manuscript.

Kind regards,

Stephan N. Witt, Ph.D.

Academic Editor

PLOS ONE

Journal Requirements:

2**.** Thank you for stating the following in the Competing Interests section:

“KSF serves as consultant and/or member of an advisory board for Novo Nordisk, Eisai/Bioarctic, Eli Lilly, Roche Diagnostics (Remuneration paid to institution) , serves (or has served) as Principal or Sub-investigator and National Coordinator on several industry-sponsored phase 2 and 3 trials (indication Alzheimer´s disease): Roche, Roche Diagnostics, Biogen, NovoNordisk, Osuka, AbbVie (Remuneration paid to institution), serves on the Scientific advisory board (and as lecturer) on the MiCog educational program (supported by an unrestricted grant from Nestlé to the European Geriatric Medicines Society, personal remuneration), speaker and educational activities for Roche, Roche Diagnostics, Eisai/Bioarctic, Eli Lilly, Novo Nordisk, Lundbeck A/S (2023, 2024, 2025) (remuneration paid to institution), BestPractice Nordic, Alzheimerforeningen, Lundbeckfonden, Folkeuni-versitetet i Emdrup, Dagens medicin (2023, 2024, 2025) (Personal remuneration), serves as Editor-in-Chief for Alzheimer´s Research & Therapy Springer – Nature (from 2024 – 2023-2024 as Associate Editor) (Personal remuneration), receives royalties from publications with Springer-Nature and Hans Reitzels Forlag and has re-ceived funding or participated in research which has received funding from the following: Alzheimer Forsknings-fonden, Parkinsonforeningen, Aase og Ejner Danielsens Fond, KID fonden, Ellen Mørch Fonden, Jascha Fond-en, C2N, Overretssagfører L. Zeuthens Mindefond, Kong Christian den Tiendes Mindefond, Rigshospitalets Forskningspulje, Innovationsfonden, A.P. Møller fonden, ERA-PERMED, IHI, Hertzfonden, Harboefonden, Grosserer F.L.Foghts Fond, Fonden for Neurologisk Forskning, DANMODIS, Beckett fonden.

Reamining authors have declared that no competing interests exist.”

3. In the online submission form, you indicated that “The datasets used and/or analysed during the current study available from the corresponding author on reasonable request”

Reviewers' comments:

Reviewer's Responses to Questions

**Comments to the Author**

1. Is the manuscript technically sound, and do the data support the conclusions?

Reviewer #1: Partly

Reviewer #2: Yes

2. Has the statistical analysis been performed appropriately and rigorously?

Reviewer #1: I Don't Know

Reviewer #2: Yes

3. Have the authors made all data underlying the findings in their manuscript fully available?

Reviewer #1: Yes

Reviewer #2: Yes

4. Is the manuscript presented in an intelligible fashion and written in standard English?

Reviewer #1: No

Reviewer #2: Yes

Reviewer #1: Bsoul et al. present a ‘universal control fluid’ and other methodological manipulations to improve the reproducibility of aSyn RT-QuIC testing when analyzing patient CSF specimens. The goals of making aSyn RT-QuIC testing simpler and more reproducible between labs are worthy ones. They showed some batch-to-batch variability amongst 6 in-house rec aSyn preps, which is to be expected based on the experience of multiple other labs. They then assessed the diagnostic performance of their rec aSyn against an 81-sample panel of CSFs from syncleinopathy and non-synucleinopathy individuals and achieved excellent results that were comparable to previous studies. As also seen in prior studies, they find that the results were influenced somewhat by the volume of CSF used per reaction. The UCF they developed was intended to mimic the matrix of CSF and to provide a well-defined and stabilizing substitute diluent for biospecimens such as CSF and brain to help reduce assay-to-assay variability.

Critique:

1) I found this manuscript imprecisely written and unusually difficult to follow: There is a confusing presentation of results in both the Methods and Results sections. This made it quite difficult to understand even the bare essentials of the experiments described in the Results without bouncing back and forth between the two sections. Even then, there was frequent use of ambiguous or under-defined/non-specific terms, and a lack of detail in Results and figure legends that required me do a lot of assuming and guessing to try figure out the comparisons actually being made in the experiments. Specific examples are noted below.

2) L224: ‘seeded RT-QuIC reactions’: >> What was the source of seed? The descriptions here and in the fig. legend suggest that the unseeded reactions received no BH at all. Is this true? Much better negative controls would be non-synucleinopathy BH. 'Unseeded' is ambiguous because it can mean either that the reaction received no biospecimen at all, or that the biospecimen it received had no seeds in it. Often the problem with certain aSyn preps is that they give some early positive reactions when seeded with negative control BH, but it seems that here only reaction mix or similar was used for the negative controls. Please clarify and justify. In any case, the reader should not have to have read the Materials and Methods in detail to comprehend the Results, or have to guess what part of the M&M applies to a given experiment.

3) L237: ‘true negative replicates’: >>Again, see previous comment, which applies here as well.

4) L239: ‘unseed’: >>ditto

5) L246-247: >>What sample set was used here (Table 3 and Fig 2)? How many of each type of CSF were tested? This not mentioned here or in the figure legend. I assume that the sample panel is the one described in Table 1, but that is not stated here. Again, the reader should not have to have read the Materials and Methods in detail to comprehend the basics of experiments presented in the Results, and have to guess what part of the M&M applies to a given experiment described in Results.

6) L280: ‘seeded’: >>With what?

7) L282: ‘good’ should be well.

8) L293: ‘BH seeded’: >>What kind of BH?

9) L296: ‘CSF samples’: >>What kind? I assume that they are synucleinopathy-positive, but this must be stated. In any case the same dilutions of non-synucleinopathy CSFs in UCF should be compared to these results.

10) L305: ‘CSF sample’: >>What kind of CSF samples??? Also explain (explicitly) the color coding of curves in (b) so the reader doesn’t have to guess whether is related to the colors in panel (a).

11) L312: ‘BH’: >>What kind?

12) L311-324 & Fig 5: >>There are problems with this experiment. One is that the UCF is causing major distortions in the baseline that are not apparent with water alone (c). It is difficult to tell whether this is due to a higher sensitivity for synucleinopathy seeds as argued by the authors, or to UCF itself, because the dilutions in (b) were not carried past the endpoint (it seems) and also (I have to assume because it isn't explained) that no non-synucleinopathy control BH was compared to whatever synucleinopathy (I assume) BH was used. Accordingly, this experiment is difficult to interpret without direct comparisons between synucleinopathy and non-synucleinopathy BHs.

13) Paragraph beginning with L329: >>I don't know what to think of the data shown in Fig. 6. UCF itself seems to have a profound effect on the fluorescence compared to the water-based measurements. Why? Importantly, the kinetics seen with UCF alone (no sample) are not shown for comparison, but they should be. Certainly, the PFFs were not diluted far enough in UCF to eliminate the 100,000+ RFU response relative to the reactions with PFF and no UCF; this would also be helpful to establish.

14) After addressing the above concerns, the Discussion and Conclusion should be reconsidered accordingly.

15) L395-6: ‘Currently, in αSynD SAA most often used negative control reactions consist of 98-100 μL 396 unseeded master mix…’: I would disagree and add that unseeded master mix is not a good control; again, a non-synucleinopathy control biospecimen should be used.

16) L398-399: ‘…unseeded and BH reactions control only for the rec αSyn amplification in the volumes of master mix added to these reactions,…’. This comparison does not control for rec aSyn amplification that occurs with a non-synucleinopathy BH specimen, and it is the latter that is by far more important diagnostically than whatever happens to rec aSyn in the absence of a matched biospecimen matrix.

Reviewer #2: Comments to authors: This is a single-site laboratory study that the establishment and validation of an αSynD RT-QuIC protocol, published by Groveman B.R. et al. The novelty of the study is that the in-houseRT-QuIC can detect and differentiate DLB and PD from a pool of different neurodegenerative diseases, including MSA, and normal controls, with 94% accuracy using 7 µL CSF, and 276 94.5% accuracy using 15 µL CSF. This study may be useful by clinicians, and health public specialists to improve patient care and health policy. The study design, conduct, and analysis were described in a manner that is unbiased, appropriate, and reproducible. The study sample was small and not neuropathologically confirmed, however, and the results are generalized. The manuscript was approved by an institutional review board

Specific comments

Feedback by section

abstract: ‘’The established RT-QuIC protocol for pathologic α-synuclein detection in CSF samples is a highly sensitive (92-96%) and specific (93-96%)’’ between which groups?

Introduction: please clarify the primary and secondary objectives of the study

Results: ‘’The CSF test was considered positive if RFU threshold was crossed in at least two out of total four sample replicates within 48 hours’’ reference here is missing?

Discussion: Other explanations that “ In case of MSA, none of the CSF samples added in 4 and 7 µL volumes yielded positive results”?

- How could advances/research that have been discussed here impact real-world outcomes (diagnosis, treatment guidelines, effectiveness, economics, drug utilization etc.)? Can changes be realistically implemented into clinical/research practice? What is preventing adoption in clinical practice?

- What are the key weaknesses/challenges in the field and how can current problems/limitations be solved? Are there any technical, technological, or methodical limitations that prevent research from advancing as it could?

- What potential does further research hold? What is the ultimate goal in this field?

- Does the future of study lie in this area? Are there other more promising areas in the field which could be progressed?

- How will the field evolve in the future? In your perspective, what will the standard procedure have gained or lost from the current norm in five or ten years?

**Do you want your identity to be public for this peer review?** For information about this choice, including consent withdrawal, please see our Privacy Policy

Reviewer #1: No

Reviewer #2: **Yes: ** Anastasia Bougea

---

## [Author Response · Author response to Decision Letter 1]

8 May 2025

Dear reviewers and editors,

Thank you for your relevant comments and suggestions to our work. We have responded to each of them, point-by-point, in the text below. As recommended, we have also performed addi-tional experiments to support the response.

In the manus copy with track changes our responses to Reviewer 1 comments are highlighted in blue, and responses to Reviewer 2 comments are highlighted in yellow.

Reviewer #1: Bsoul et al. present a ‘universal control fluid’ and other methodological manipula-tions to improve the reproducibility of aSyn RT-QuIC testing when analyzing patient CSF spec-imens. The goals of making aSyn RT-QuIC testing simpler and more reproducible between labs are worthy ones. They showed some batch-to-batch variability amongst 6 in-house rec aSyn preps, which is to be expected based on the experience of multiple other labs. They then as-sessed the diagnostic performance of their rec aSyn against an 81-sample panel of CSFs from syncleinopathy and non-synucleinopathy individuals and achieved excellent results that were comparable to previous studies. As also seen in prior studies, they find that the results were in-fluenced somewhat by the volume of CSF used per reaction. The UCF they developed was in-tended to mimic the matrix of CSF and to provide a well-defined and stabilizing substitute dilu-ent for biospecimens such as CSF and brain to help reduce assay-to-assay variability.

Comments:

1) I found this manuscript imprecisely written and unusually difficult to follow: There is a con-fusing presentation of results in both the Methods and Results sections. This made it quite diffi-cult to understand even the bare essentials of the experiments described in the Results without bouncing back and forth between the two sections. Even then, there was frequent use of ambig-uous or under-defined/non-specific terms, and a lack of detail in Results and figure legends that required me do a lot of assuming and guessing to try figure out the comparisons actually being made in the experiments. Specific examples are noted below.

Response: We appreciate the specific examples listed below, and we have made changes to the text with focus on specifying the indicated ambiguous terms and explaining the results.

2) L224: ‘seeded RT-QuIC reactions’: >> What was the source of seed? The descriptions here and in the fig. legend suggest that the unseeded reactions received no BH at all. Is this true? Much better negative controls would be non-synucleinopathy BH. 'Unseeded' is ambiguous be-cause it can mean either that the reaction received no biospecimen at all, or that the biospecimen it received had no seeds in it. Often the problem with certain aSyn preps is that they give some early positive reactions when seeded with negative control BH, but it seems that here only reac-tion mix or similar was used for the negative controls. Please clarify and justify. In any case, the reader should not have to have read the Materials and Methods in detail to comprehend the Re-sults, or have to guess what part of the M&M applies to a given experiment.

Response: We used the term “unseeded” to indicate that reactions’ master mix was used in its neat composition, i.e. not seeded with either DLB or normal brain homogenates. We have now clarified this when introducing the term “unseeded” in L146 and “seeded” in L147 in M&M section.

Our reasoning for using neat master mix as the negative control is 1) to track any changes in the performance of the recombinant aSyn itself, unbiased by added BH and CSF, which, with their unique, patient-specific, compositions can suppress or trigger recombinant aSyn aggregation, and thus, over time, lead to wrong conclusions; and 2) to move away from using patient biological samples as controls in order to uniform and simplify the assay.

We have performed an additional experiment to test the performance of our establish aSyn SAA with reactions containing 2uL normal brain homogenate (diluted in water from 10-3 to 10-6, each dilution with 22 replicated) to 89uL master mix. All assay conditions were kept the same. Re-sults indicated no aggregation in any of the replicates. This has been briefly indicated in the M&M section.

3) L237: ‘true negative replicates’: >>Again, see previous comment, which applies here as well.

Response: We have removed “true” to avoid the confusion.

4) L239: ‘unseed’: >>ditto

Response: We have clarified the terms in this paragraph.

5) L246-247: >>What sample set was used here (Table 3 and Fig 2)? How many of each type of CSF were tested? This not mentioned here or in the figure legend. I assume that the sample panel is the one described in Table 1, but that is not stated here. Again, the reader should not have to have read the Materials and Methods in detail to comprehend the basics of experiments present-ed in the Results, and have to guess what part of the M&M applies to a given experiment de-scribed in Results.

Response: We have referred to table 1 when introducing these results.

6) L280: ‘seeded’: >>With what?

Response: We have clarified the terms in this paragraph.

7) L282: ‘good’ should be well.

Response: Thank you. That is corrected.

8) L293: ‘BH seeded’: >>What kind of BH?

Response: This has been clarified.

9) L296: ‘CSF samples’: >>What kind? I assume that they are synucleinopathy-positive, but this must be stated. In any case the same dilutions of non-synucleinopathy CSFs in UCF should be compared to these results.

Response: It has been clarified that the CSF samples are from the PD/DLB CSF cohort (Table 1) and were αSynD SAA positive.

As recommended, to evaluate the UCF performance with the αSynD SAA negative CSF samples from the cohort (7 samples total), we tested 2 uL, 4 uL, and 6 uL of normal CSF mixed with 5 uL, 3 uL and 1 uL UCF, respectively. All CSF/UCF ratio combinations for each sample were run in quadruplicate. Results indicated no false positive or higher aggregation rate in any of the rep-licates. Thus indicating no difference in UCF performance with αSynD SAA positive and nega-tive CSF samples. This has been mentioned in the M&M, Results.

10) L305: ‘CSF sample’: >>What kind of CSF samples??? Also explain (explicitly) the color coding of curves in (b) so the reader doesn’t have to guess whether is related to the colors in panel (a).

Response: It has been clarified that the CSF samples are from the PD/DLB CSF cohort (table 1) and were aSyn SAA positive. The color coding has been explained in the figure text.

11) L312: ‘BH’: >>What kind?

Response: This has been clarified.

12) L311-324 & Fig 5: >>There are problems with this experiment. One is that the UCF is caus-ing major distortions in the baseline that are not apparent with water alone (c). It is difficult to tell whether this is due to a higher sensitivity for synucleinopathy seeds as argued by the authors, or to UCF itself, because the dilutions in (b) were not carried past the endpoint (it seems) and also (I have to assume because it isn't explained) that no non-synucleinopathy control BH was compared to whatever synucleinopathy (I assume) BH was used. Accordingly, this experiment is difficult to interpret without direct comparisons between synucleinopathy and non-synucleinopathy BHs.

Response: The aim of this experiment is to compare currently used SAA positive control set-up, the DLB BH diluted in water set-up, with the proposed DLB BH diluted in UCF set-up. We did not define the endpoint of DLB BH dilutions in UCF, however, seeing a clear linear regression and smooth kinetic patterns reflecting dilution factor, we can assume that with a few additional dilutions, the RFU threshold would not be crossed. We did not test the non-synucleinopathy BHs in this experiment, because this set-up is not used in our SAA protocol, and that was not the aim of this experiment. Furthermore, we do not see any UCF induced rec aSyn aggregation in reac-tions with healthy patient CSF. On the contrary, the data indicate that UCF have reaction stabiliz-ing effect.

We have revised this section with clarification of terms used.

13) Paragraph beginning with L329: >>I don't know what to think of the data shown in Fig. 6. UCF itself seems to have a profound effect on the fluorescence compared to the water-based measurements. Why? Importantly, the kinetics seen with UCF alone (no sample) are not shown for comparison, but they should be. Certainly, the PFFs were not diluted far enough in UCF to eliminate the 100,000+ RFU response relative to the reactions with PFF and no UCF; this would also be helpful to establish.

Response: The aim of this experiment is to evaluate if PFF diluted in UCF demonstrate linear regression between PFF concentration and RFU, and if the kinetics within the set range of dilu-tions demonstrate uniform curves, indicating that the reactions with UCF as a diluent are stable and predictable. From other experiments described in the paper (i.e., Fig 3, Fig 4b), we know that reactions with UCF alone do not induce rec aSyn aggregation.

14) After addressing the above concerns, the Discussion and Conclusion should be reconsidered accordingly.

Response: We have updated discussion and conclusion accordingly.

15) L395-6: ‘Currently, in αSynD SAA most often used negative control reactions consist of 98-100 μL 396 unseeded master mix…’: I would disagree and add that unseeded master mix is not a good control; again, a non-synucleinopathy control biospecimen should be used.

Response: We have acknowledged that and indicate that we do not see a difference in the per-formance of the 2 negative control set-ups (based on data from the experiment mentioned under point 2).

16) L398-399: ‘…unseeded and BH reactions control only for the rec αSyn amplification in the volumes of master mix added to these reactions,…’. This comparison does not control for rec aSyn amplification that occurs with a non-synucleinopathy BH specimen, and it is the latter that is by far more important diagnostically than whatever happens to rec aSyn in the absence of a matched biospecimen matrix.

Response: We consider the unseeded (with or without BH) and seeded reactions to primarily track the performance of the assay (i.e. technical reproducibility) but not the diagnostic accuracy of CSF testing. That is due to the inherent difference between reactions with added 2 uL BH, and 7 uL or 15 uL of CSF. We have rephrased this section to focus our discussion on the differ-ences between current standard controls and CSF testing, because the primary point is to intro-duce an alternative control set-up based on UCF.

Reviewer #2: Comments to authors: This is a single-site laboratory study that the establishment and validation of an αSynD RT-QuIC protocol, published by Groveman B.R. et al. The novelty of the study is that the in-houseRT-QuIC can detect and differentiate DLB and PD from a pool of different neurodegenerative diseases, including MSA, and normal controls, with 94% accura-cy using 7 µL CSF, and 276 94.5% accuracy using 15 µL CSF. This study may be useful by clinicians, and health public specialists to improve patient care and health policy. The study de-sign, conduct, and analysis were described in a manner that is unbiased, appropriate, and repro-ducible. The study sample was small and not neuropathologically confirmed, however, and the results are generalized. The manuscript was approved by an institutional review board

Specific comments

Feedback by section

abstract: ‘’The established RT-QuIC protocol for pathologic α-synuclein detection in CSF sam-ples is a highly sensitive (92-96%) and specific (93-96%)’’ between which groups?

Response: It has been clarified that the reported sensitivity and specificity is regarding PD/DLB.

Introduction: please clarify the primary and secondary objectives of the study

Response: The primary and secondary objective has been clarified.

Results: ‘’The CSF test was considered positive if RFU threshold was crossed in at least two out of total four sample replicates within 48 hours’’ reference here is missing?

Response: Added reference 16 and 12.

Discussion: Other explanations that “ In case of MSA, none of the CSF samples added in 4 and 7 µL volumes yielded positive results”?

Response: We believe the explanation is the technical assay parameters and the type of rec aSyn used in our protocol that are not compatible with MSA aSyn strain detection.

- How could advances/research that have been discussed here impact real-world outcomes (di-agnosis, treatment guidelines, effectiveness, economics, drug utilization etc.)? Can changes be realistically implemented into clinical/research practice? What is preventing adoption in clinical practice?

Response: Please see added text in the discussion.

- What are the key weaknesses/challenges in the field and how can current problems/limitations be solved? Are there any technical, technological, or methodical limitations that prevent research from advancing as it could?

Response: Please see added text in the discussion. To support the answer and for more elaborate review of SAA limitations and prospects we added 2 new references (ref 23 and 32)

- What potential does further research hold? What is the ultimate goal in this field?

Response: Please see added text in the discussion.

- Does the future of study lie in this area? Are there other more promising areas in the field which could be progressed?

Response: Please see added text in the discussion and further elaboration on the topic in refer-ences 23 and 32.

- How will the field evolve in the future? In your perspective, what will the standard procedure have gained or lost from the current norm in five or ten years?

Response: Please see added text in the discussion and conclusions.

---

## [Decision Letter · Decision Letter 1]

Seeding Amplification Assay with Universal Control Fluid: Standardized Detection of α-Synucleinopathies

PONE-D-25-09326R1

Dear Dr.  Areškevičiūtė,

We’re pleased to inform you that your manuscript has been judged scientifically suitable for publication and will be formally accepted for publication once it meets all outstanding technical requirements.

Kind regards,

Stephan N. Witt, Ph.D.

Academic Editor

PLOS ONE

Additional Editor Comments (optional):

Reviewers' comments:

Reviewer's Responses to Questions

**Comments to the Author**

Reviewer #1: All comments have been addressed

2. Is the manuscript technically sound, and do the data support the conclusions?

Reviewer #1: Yes

3. Has the statistical analysis been performed appropriately and rigorously?

Reviewer #1: Yes

4. Have the authors made all data underlying the findings in their manuscript fully available?

Reviewer #1: Yes

5. Is the manuscript presented in an intelligible fashion and written in standard English?

Reviewer #1: Yes

Reviewer #1: The authors have addressed my concerns reasonably................................................................

**Do you want your identity to be public for this peer review?** For information about this choice, including consent withdrawal, please see our Privacy Policy

Reviewer #1: No

---

## [Editor Report · Acceptance letter]

PONE-D-25-09326R1

PLOS ONE

Dear Dr. Areškevičiūtė,

I'm pleased to inform you that your manuscript has been deemed suitable for publication in PLOS ONE. Congratulations! Your manuscript is now being handed over to our production team.

Kind regards,

on behalf of

Dr. Stephan N. Witt

Academic Editor

PLOS ONE